Involvement of the arachidonic acid cytochrome P450 epoxygenase pathway in the proliferation and invasion of human multiple myeloma cells

Shao Jing 1 2
Wang Hongxiang 2
Yuan Guolin 3
Chen Zhichao 1
Li Qiubai 1 qiubaili@hust.edu.cn
1 Institute of Hematology, Union Hospital, Tongji Medical College, Huazhong University of Science and Technology , Wuhan Hubei , China
2 Wuhan Central Hospital, Department of Hematology , Wuhan Hubei , China
3 Xiangyang Central Hospital, the Affiliated Hospital of Hubei University of Arts and Science, Department of Hematology , Xiangyang Hubei , China
Kim Cheorl-Ho
Electronic publication date: 2016 Apr 11
Publication date: 2016
Volume: 4
Electronic Location ID: e1925
Received 2015 Dec 9; Accepted 2016 Mar 22
Copyright: ©2016 Shao et al.
Copyright year: 2016
Copyright holder: Shao et al.
License: This is an open access article distributed under the terms of the Creative Commons Attribution License, which permits unrestricted use, distribution, reproduction and adaptation in any medium and for any purpose provided that it is properly attributed. For attribution, the original author(s), title, publication source (PeerJ) and either DOI or URL of the article must be cited.
License URL: https://creativecommons.org/licenses/by/4.0/

Keywords: Multiple myeloma, Proliferation, Epoxyeicosatrienoic acids, Cytochrome p450, Apoptosis, Migration, Invasion

Funding: National Natural Science Foundation of China 81272624 and 81071943 This work was supported by grants from the National Natural Science Foundation of China to Qiubai Li (No. 81272624 and No. 81071943). The funders had no role in study design, data collection and analysis, decision to publish, or preparation of the manuscript.

==============================
Cytochrome P450 (CYP) epoxygenases and the metabolites epoxyeicosatrienoic acids (EETs) exert multiple biological effects in various malignancies. We have previously found EETs to be secreted by multiple myeloma (MM) cells and to be involved in MM angiogenesis, but the role of the arachidonic acid cytochrome P450 epoxygenase pathway in the proliferation and mobility of MM cells remains unknown. In the present study, we found that MM cell lines generated detectable levels of 11,12-EET/14,15-EET and that increased levels of EETs were found in the serum of MM patients compared to healthy donors. The addition of exogenous EETs induced significantly enhanced proliferation of MM cells, whereas 17-octadecynoic acid (17-ODYA), an inhibitor of the CYP epoxygenase pathway, inhibited the viability and proliferation of MM cells. Moreover, this inhibitory effect could be successfully reversed by exogenous EETs. 17-ODYA also inhibited the motility of MM cells in a time-dependent manner, with a reduction of the gelatinolytic activity and protein expression of the matrix metalloproteinases (MMP)-2 and MMP-9. These results suggest the CYP epoxygenase pathway to be involved in the proliferation and invasion of MM cells, for which 17-ODYA could be a promising therapeutic drug.

Introduction

It is well known that arachidonic acid (AA) is converted to eicosanoid mediators by the cyclooxygenases (COX), lipoxygenases (LOX), and cytochrome P450 (CYP) epoxygenase pathways to generate hundreds of metabolites that have different biological activities and contribute to the pathogenesis of numerous diseases. The changes in the expression level and distribution pattern of enzymes involved in eicosanoid biosynthesis may be especially relevant in carcinogenesis (Koch et al., 2011; Wasilewicz et al., 2010). Multiple myeloma (MM) is an incurable plasma cell malignancy characterized by the accumulation of long-living plasma cells in the bone marrow. Current evidence supports the notion of an association between deranged endogenous fatty acids and MM pathogenesis, while the abnormal fatty acid profile may contribute to cancer-associated inflammation through an abnormal arachidonic acids metabolism (Jurczyszyn et al., 2014; Jurczyszyn et al., 2015). AA metabolism pathways are also involved in the pathogenesis of multiple myeloma (MM). COX-2 overexpression is associated with reduced estimated survival of MM patients and unfavorable prognostic factors such as LDH, age, and β2-microglobulin (Cetin et al., 2005). 12-LOX is also detected in MM cells, and the inhibitor Baicalein suppresses the proliferation of MM cells (Li et al., 2006). However, the role of the cytochrome P450 epoxygenase pathway in the disease progression of MM remains poorly understood.

Primary metabolic products of CYP epoxygenases include four epoxyeicosatrienoic acid (EET) regioisomers: 5,6-EET, 8,9-EET, 11,12-EET, and 14,15-EET. Upon hydration by soluble epoxide hydrolase (sEH), EETs are converted to more stable and less biologically active metabolites, dihydroxyeicosatrienoic acids (DHETs) (Capdevila et al., 1992; VanRollins et al., 1993). Although each purified CYP epoxygenase converts arachidonic acid to all four EET regioisomers, the main products in many cases are 11,12-EET and 14,15-EET (Capdevila, Falck & Harris, 2000). Studies have shown that small amounts of 8,9-, 11,12-, and 14,15-EETs are present in the plasma, the liver, and the kidney, with 14,15-EET being the most abundant regioisomer (Karara et al., 1991; Karara et al., 1990). EETs are autocrine and paracrine mediators that function primarily in the cardiovascular and renal system, and play a key role in inflammation and tissue homeostasis in the vascular system (Campbell & Harder, 1999). Recently, there is increasing evidence suggesting potential roles of CYP epoxygenases and EETs in tumors (Chen et al., 2011; Jiang et al., 2005; Nithipatikom et al., 2010). EETs dramatically enhance the proliferation and motility of tumor cells, and overexpression of CYP epoxygenases has the same effects in tumor cells in vitro and in vivo.

We previously demonstrated that MM cells secrete EETs into the supernatant and that the CYP epoxygenase pathway participates in the MM cell-induced angiogenesis, which can be inhibited by 17-octadecynoic acid (17-ODYA), an inhibitor of CYP epoxygenase pathway (Shao et al., 2011). In this study, we further investigated the role of the CYP epoxygenase pathway in cell proliferation, apoptosis, migration and invasion of MM.

Materials and Methods

Materials

Cell culture medium (RPMI 1640 medium) and fetal bovine serum (FBS) were purchased from Invitrogen (Carlsbad, CA, USA). The ELISA kits for 11,12-EET and 14,15-EET were obtained from Detroit R&D (Detroit, MI, USA). 11,12-EET, 14,15-EET, and 17-ODYA were purchased from Cayman Chemical (Ann Arbor, MI, USA). 3-(4, 5-dimethyl-2-thiazy1)-2,5-diphenyl-2H-tetrazolium bromide (MTT) and DMSO were obtained from Sigma Chemical Co. (St. Louis, MO, USA). PI and Annexin V/FITC detection kits were purchased from Bender Medsystems Inc. (Burlingame, CA, USA). Antibodies against CyclinD1, Bax, Bcl-2, MMP-2, and MMP-9 were purchased from Epitomics Inc. (Burlingame, CA, USA). Horseradish peroxidase (HRP)-conjugated secondary antibodies (goat-anti-rabbit IgG) were purchased from KPL (Gaithersburg, MA, USA). Enhanced chemiluminescence reagents were purchased from Pierce, Inc. (Rockford, IL, USA). The Transwell plates were obtained from Corning Costar (Cambridge, MA, USA), and the Matrigel was purchased from BD Biosciences (Bedford, MA, USA).

Patient samples

The human study was approved by the Institutional Review Board (Review Board of Tongji Medical College, Huazhong University of Science and Technology, No 2012S119). Sixteen patients diagnosed with MM were selected for the present study after providing informed consent. Three healthy subjects were recruited as controls. Six ml of peripheral blood was collected from all cases. Serum was isolated from peripheral blood by centrifugation at 2,000 rpm for 10 min and was frozen at -80 ∘C for the EET Elisa assay. Twelve patients were men, and four patients were women (mean age 51 ± 10 years, range 34–66). Fifteen patients were diagnosed as stage III MM, and 1 patient was stage II. Nine patients were group A, and seven were group B. Eight patients were newly diagnosed, and three patients were relapse cases (Table 1).

Table 1 Clinical features of patients with multiple myeloma.

Patient no.	Age/Sex	Durie-Salmon stage	Paraprotein type	β2 − M(mg/L)	LDH(U/L)	Cytogenetic feature	
1	58/F	stageIII a	IgG, λ	4.9	220	1q21	
2	48/M	stageIIIa	IgD, λ	18	164	14q32;13q14.3;1q21	
3	62/M	stageIIIa	IgG, κ	2.9	NA	13q14.3	
4	66/M	stageIIIa	IgD, λ	35.5	188	Negative	
5	47/M	stageIII	BJP, κ	12.6	205	NA	
6	39/M	stageIII	IgD, λ	4.5	201	14q32	
7	60/M	stageIIIa	IgG, λ	15.1	177	17p13.1	
8	60/M	stageIIa	IgD, λ	3.3	257	Negative	
9	35/M	stageIIIa	Nonsecreting type	11.3	177	13q14	
10	46/M	stageIII	IgA, κ	3.5	263	NA	
11	58/F	stageIII	BJP, κ	1.3	200	NA	
12	60/F	stageIII	IgA, λ	2.8	NA	14q32	
13	34/F	stageIIIa	Nonsecreting type	9.3	159	Negative	
14	40/M	stageIII	IgA, λ	1.5	234	17p13.1/17p11.1-q11.1	
15	53/M	stageIII	IgG, λ	1.8	106	14q32;13q14	
16	44/M	stageIII	BJP, λ	10.1	176	Negative	
Notes.

AbbreviationsM male

F female

Ig immunoglobulin

BJP Bence Jones protein

β2-M β2-microglobulin

NA not available

a Newly diagnosed.

Cell lines and cell culture

Multiple myeloma cell lines U266 and RPMI8226 were obtained from the American Type Culture Collection (ATCC, Rockville, MD, USA). U266 and RPMI 8226 cells were cultured in RPMI 1640 medium with 10% fetal bovine serum (FBS), 100 units/ml penicillin and 100 µg/ml streptomycin at 37 ∘C in a humidified 95% air/ 5% CO2 atmosphere incubator.

EET ELISA

For the measurement of EETs in MM cells, cells (1 × 106/ml) were plated in 6-well plates in the presence or absence of 17-ODYA (100 µmol/L). Cells were collected by centrifugation at 2,000 rpm for 10 min. EETs in the MM cells and serum of multiple myeloma patients were determined using an ELISA kit according to the manufacturer’s instructions.

Cell viability and proliferation assays

MM cells were synchronized by incubation in FBS-free RPMI 1640 overnight. 11,12-EET, 14,15-EET and 17-ODYA were added at various concentrations to the media. The cells were cultured for 24, 48 or 72 h. Because of the instability of EETs, they were added to the media every 4–6 h. DMSO was used as a vehicle for all compounds. After treatment with MTT (20 µl per well) for 4 h, DMSO was added to dissolve the crystalline reaction products. The plates were read at a wavelength of 490 nm. Each group measurement was repeated in five duplicate wells.

Determination of cell apoptosis and cell cycle by flow cytometry

Cells were synchronized by incubation in FBS-free RPMI 1640 overnight. These cells (1×106) were plated in 6-well plates and treated with 17-ODYA at 100 µmol/L for 48 h. Subsequently, the cells were harvested and incubated with FITC-conjugated Annexin V and propidium iodide, according to the manufacturer’s protocol, and analyzed using a flow cytometer (FACScan, Becton Dickinson, USA).

To analyze the cell cycle distribution, cells (1×106) were cultured with 17-ODYA (100 µmol/L) for 48 h. Thereafter, the cells were harvested and fixed with 75% ethanol at −20∘C overnight. After removing the ethanol, the cells were incubated with RNase (250 µg/ml) at 37 ∘C for 30 min followed by propidium iodide (50 µg/ml) at 4 ∘C for 30 min. Cell cycle analysis was determined by flow cytometry.

Western blotting

Proteins from the cell lysates were separated by SDS-PAGE electrophoresis and transferred to a nitrocellulose filter membrane. The membrane was blocked with 5% nonfat milk and incubated with an antibody against a specific protein antigen. The location of the antibody-antigen complex on the Western blot was revealed by incubation with a peroxidase-conjugated secondary antibody. The bands were visualized using the enhanced chemiluminescence method.

Migration assay

Migration assays were conducted as previously described in detail (Fuchida et al., 2008). Briefly, the assay was performed using a 24-well Transwell plates with a polycarbonate membrane with a pore size of 8 µm. Before the migration assay, cells were treated with 17-ODYA (100 µmol/L) for 24 or 48 h. The cells were centrifuged and suspended in serum-free RPMI 1640 at a density of 1×106/ml. Volumes of 1×105 cells were seeded into the upper chambers, and other cells were cultured concurrently in the mother liquor to calculate the migration ratio. The lower chamber contained RPMI 1640 with 10% FBS as a chemoattractant. After 24 h of incubation at 37 ∘C, the numbers of cells transmigrating into the lower chamber and in the mother liquor were counted using a flow cytometer (FACScan) gated for 20 s at a high flow rate. The experiments were performed in triplicate. The migration ratio was calculated as the migrated cells as the percentage of cells in the mother liquor.

Invasion assay

The invasion assay was performed as the migration assay, except for the fact that the upper surface of the polycarbonate membranes was coated with Matrigel and dried overnight at 37 ∘C. The Transwell plates were incubated at 37 ∘C for 24 or 48 h. The cell number was counted by flow cytometry. The invasion ratio was calculated as the migrated cells as the percentage of cells in the mother liquor.

Gelatin zymography

Gelatin zymography was performed using the supernatant of cultured MM cells to evaluate the activity of secreted MMP-2 and MMP-9. Volumes of 1 × 106 cells were incubated with or without 17-ODYA (100 µmol/L) for 24 h. The culture supernatants were collected and centrifuged at 2000 rpm for 10 min. Supernatants containing the same amount of proteins from each group were applied to 10% SDS-PAGE (containing 0.1% gelatin). After electrophoresis, the gels were washed in 2.5% Triton X-100 for 80 min to remove SDS. The gels were incubated for 20 h at 37 ∘C and stained with 0.5% Coomassie brilliant blue for 3 h. After staining, the gels were destained with 30% methanol and 10% acetic acid.

Statistics

The data are presented as the mean ± SE. All experiments were done with triplicate replications of each treatment group (n = 3). Student’s t test or ANOVA was performed as appropriate to determine the statistical significance of differences among different groups. The Mann–Whitney two-sample test was used to investigate the significant differences of EET concentration between MM patients and healthy controls. Correlation analysis was used to examine the correlation between sets of variables in MM patients. In all cases, statistical significance was defined by P <0.05.

Results

EETs in MM cell lines and the peripheral blood serum of MM patients

We previously found secreted 11,12-EET and 14,15-EET in the supernatant of MM cell lines (Shao et al., 2011), but the expression levels of 11,12-EET and 14,15-EET in MM cells is unknown. Therefore, we directly assessed the levels of these EETs in MM cells. Based upon the results of EET ELISA assay, we found both MM cell lines U266 and RPMI 8226 produced 11,12-EET and 14,15-EET (Fig. 1A). The levels of 14,15-EET were much higher than those of 11,12-EET in the two cell lines, suggesting that 14,15-EET was the most abundant regioisomer (Karara et al., 1991; Karara et al., 1990). We also evaluated the levels of EETs in MM patients after collecting the peripheral blood serum of 16 patients and three healthy volunteers, as shown in Table 1. The results showed that the concentrations of 11,12-EET and 14,15-EET were significantly higher in patients than in healthy donors (Fig. 1B). The mean concentration of 11,12-EET in patient serum was 291.94 ± 383.98 ng/ml (range from 0.78 to 1193.36 ng/ml), which was significantly different from the control (5.10 ±2.31 ng/ml, range from 2.86 to 7.47 ng/ml,) (P < 0.01). The mean concentration of 14,15-EET in MM serum was 1056.48 ± 906.47 ng/ml (range from 0.61 to 2754.99 ng/ml), which was markedly higher than the control with 4.92 ± 2.32 ng/ml (range from 2.54 to 7.18 ng/ml) (P < 0.01). Meanwhile, we analyzed the correlations of EET concentrations with the prognostic factors, such as LDH and β2-microglobulin, but no statistically significant correlation was found. With regard to LDH and β2-microglobulin, the results for 11,12-EET were r = −0.139, P > 0.05 and r = − 0.27, P > 0.05, respectively, and for 14,15-EET, r = − 0.395, P > 0.05 and r = − 0.114, P > 0.05, respectively.

Figure 1 Levels of EETs in MM cells and serum of MM patients.

11,12-EET and 14,15-EET levels were determined by ELISA following the instruction of manufacturers. (A) MM cell lines (1×106 cells) expressed 11, 12-EET and 14, 15-EET. (B) 11, 12-EET and 14, 15-EET levels from healthy volunteers and patients with multiple myeloma (the ordinate was for log transformation; EETs in MM patients serum were significantly higher compared with healthy subjects, P < 0.01).

Figure 2 17-ODYA decreases EETs level in MM cells and suppresses MM cell viability.

(A) once addition of exogenous 11,12-EET or 14,15-EET increased the viability of U266 and RPMI8226 (∗P < 0.05 versus DMSO vehicle group). (B) 17-ODYA (100 µmol/L) treated U266 and RPMI8226 for 24 h or 48 h, and Elisa assay was used to detect the 11, 12-EET and 14, 15-EET levels in MM cells (∗P < 0.05 versus control). (C) 17-ODYA (100 μmol/L) decreased cell viability of U266 and RPMI8226 in dose and time dependent manners (∗P < 0.05 versus control). (D) 17-ODYA (100 μmol/L) suppressed the proliferation of U266 and RPMI8226 cells, which could be reversed by exogenous EETs (all groups were treated with 100 μmol/L 17-ODYA for 24 h; ∗P < 0.05 versus vehicle group).

17-ODYA suppressed EET levels and the proliferation of MM cells

Based on previous studies (Chen et al., 2011; Jiang et al., 2005; Nithipatikom et al., 2010) and our above results, we hypothesized that EETs may contribute to the neoplastic phenotype of MM. First, we determined the effect of exogenous EETs on the proliferation of MM cells and found that concentrations of 11,12-EET and 14,15-EET in the range of 100 nmol/L to 400 nmol/L increased the proliferation of U266 and RPMI 8226 cells in vitro for 24, 48 and 72 h (Fig. 2A). These EEts also promoted cell viability in a dose and time dependent-manners. Because of the instability of EETs, they should be added to the medium every 4–6 h, but the vehicle (DMSO) significantly affected the viability of the cells (Fig. S1). Thus, we analyzed the differences between the vehicle and experimental groups. As 17-ODYA is the inhibitor of CYP epoxygenases, we measured the levels of EETs in MM cells in the presence or absence of 17-ODYA to further characterize the effect of 17-ODYA on the biosynthesis of eicosanoids in MM cells. Our findings revealed that 17-ODYA decreased both the levels of 11,12-EET and 14,15-EET in U266 and RPMI 8226 cells (Fig. 2B). Meanwhile, the addition of the epoxygenase inhibitor 17-ODYA decreased the proliferation of MM cells, and the inhibition ratio increased with the dose and duration of treatment (Fig. 2C). Importantly, exogenous EETs reversed the 17-ODYA-mediated decrease in proliferation of the MM cell lines compared to the vehicle (DMSO) group (Fig. 2D). These results indicate that CYP epoxygenase-derived EETs promote the viability of MM cells and that CYP epoxygenases may play an important role in the proliferation of multiple myeloma.

17-ODYA induced apoptosis and cell cycle arrest of MM cells

We next investigated whether 17-ODYA induced cellular apoptosis or disrupted the cell cycle. After treatment with 17-ODYA for 48 h, the ratios of apoptotic cells were assessed using Annexin V/PI staining. Treatment of both U266 and RPMI 8226 cells with 100 µmol/L 17-ODYA increased apoptosis compared with control (blank) group (P < 0.05; Fig. 3A). Treatment with 17-ODYA (100 µmol/L) for 48 h also increased the percentage of cells in G0/G1 phase in both MM cell lines compared to the control (blank) and vehicle (DMSO) group (P < 0.05; Fig. 3B). We next measured apoptosis and cell cycle-related proteins by Western blot analysis, and the results revealed that treatment with 17-ODYA decreased the levels of the G1 phase regulatory protein cyclin D1 and antiapoptotic protein Bcl-2, while increasing the level of proapoptotic protein Bax (Fig. 4). Collectively, these results suggest that the epoxygenase pathway is possibly involved in enhancing the proliferation of myeloma cells by protecting cells from apoptosis and promoting the cell cycle progression.

17-ODYA inhibited the motility of MM cells

Migration and invasion of MM cells were evaluated using a Transwell model in vitro. 17-ODYA significantly decreased cell migration of both MM cell lines, and the migration ratio decreased in a time-dependent manner (Fig. 5A). The migration ratio of 17-ODYA treatment for 24 h was 0.93 ± 0.07% compared to 1.37 ± 0.13% for the control (blank) group (P < 0.05), and 0.10 ± 0.02% for the 48 h group compared with the control (blank) group (P < 0.05) in U266 cells. The results of RPMI 8226 cells were 1.07 ± 0.36% for the 24 h group and 0.48 ± 0.04% for the 48 h group compared to 1.57 ± 0.06% for the control (blank) group (P < 0.05). 17-ODYA also inhibited the invasion of MM cells (Fig. 5B). As MMPs play an important role in tumor metastasis, we used gelatin zymography assays and Western blotting to investigate the activity and protein content of MMPs in MM cells. As shown in fig. 5C, 17-ODYA treatment suppressed the activity of MMPs in the supernatant. To investigate whether the 17-ODYA-induced inhibition of MMP-2 and MMP-9 activity was caused by changes in the protein levels, MM cells were also analyzed by Western blot. The results showed that the levels of MMP-2 and MMP-9 were decreased by 17-ODYA treatment in both MM cell lines (Fig. 5D). Altogether, the inhibitor 17-ODYA suppressed the motility of MM cells and reduced the activity and protein levels of secreted MMPs.

Figure 3 Epoxygenase inhibitor 17-ODYA enhances MM cell apoptosis and induces cell arrest at G0/G1 phase.

(A) apoptosis of MM cells was increased by 17-ODYA (100 μmol/L) treatment for 48 h (∗P < 0.05 versus control); (B) cell arrest at G0/G1 phase was induced by 17-ODYA (100 μmol/L) treatment for 48 h (∗P < 0.05 versus control).

Figure 4 Effect of17-ODYA on levels of Bax, Bcl-2 and cyclin D1 protein in MM.

Treatment of both U266 and RPMI 8226 cells with 17-ODYA at 100 μmol/L for 48 h decreased levels of cyclin D1, the G1 phase regulatory protein, and antiapoptotic protein Bcl-2, while increased the level of Bax, a proapoptotic protein.

Figure 5 17-ODYA inhibits MM cell mobility through reducing MMP activity and protein levels.

MM cells were treated by 17-ODYA (100 μmol/L) for 24 h or 48 h, and DMSO was used for the vehicle. (A) 17-ODYA inhibited the migration of MM cells (∗P > 0.05 versus control; ∗∗P < 0.05 versus control). (B) 17-ODYA inhibited the invasion of MM cells (∗P > 0.05 versus control; ∗∗P < 0.05 versus control). (C) 17-ODYA reduced MMPs activity in the supernatant of MM cells (∗P < 0.05 versus control). (D) 17-ODYA reduced MMPs protein level in MM cells (∗P < 0.05 versus control).

Discussion

As the main products of CYP epoxygenases in many MM cases are 11,12-EET and 14,15-EET (Capdevila, Falck & Harris, 2000), we measured these two EETs in MM cells and in the serum of MM patients. In the present study, we found that both MM cell lines (U266, RPMI 8226) produced different levels of 11,12-EET and 14,15-EET and that EETs were also detected in high concentrations in the serum of MM patients. Accordingly, in leukemia and lymphoma, two other hematological malignancies, high levels of EETs in the urine and blood of patients compared to the healthy controls were reported by (Chen et al., 2011). These findings indicate that high levels of the EETs are one of the pathological characteristics of hematological malignancies, including MM. The clinical role of EETs in MM stratification and prognosis is to be determined in future studies.

EETs are the major biologically active metabolites of CYP epoxygenases and are locally active small molecule lipid mediators that play a central role in various cellular functions, including proliferation, migration and angiogenesis (Chen, Capdevila & Harris, 2001; Enayetallah, French & Grant, 2006; Jiang et al., 2005; Michaelis et al., 2005; Node et al., 1999; Wei et al., 2014). In this study, we found that both 11,12-EET and 14,15-EET significantly promoted the proliferation of MM cells, while the inhibitor 17-ODYA suppressed EET levels and the viability of all tested MM cells in a dose- and time-dependent manner. Additionally, the 17-ODYA-mediated decrease of proliferation was reversed by exogenous EETs. These findings support the notion that the elevation of EETs promotes the viability of MM cells. To be noted, 17-ODYA is also considered as an inhibitor of cytochrome P-450 omega-hydroxylase (Ohata et al., 2010), but 20-HETE, the product of cytochrome P-450 omega-hydroxylase, was not considered in the present study. Although the expression and role of 20-HETE is unknown in MM cells, it is important to be further demonstrated using other specific epoxygenase inhibitors, such as MS-PPOH (Wang et al., 1998).

In the present study, 17-ODYA increased the apoptosis and induced cell cycle arrest at the G0/G1 phase in MM cells, suggesting that 17-ODYA-induced suppression of MM cells proliferation is possibly mediated through the induction of apoptosis and cell cycle arrest. Our results also showed that the G1 phase regulatory protein cyclin D1 and Bcl-2 antiapoptotic protein were markedly down-regulated by 17-ODYA, along with up-regulation of Bax proapoptotic protein. These proteins have been shown to play a key role in MM pathogenesis (Marsaud et al., 2010). Thus, together with our previous findings about EET-induced angiogenesis in MM, the present study supported the hypothesis that EETs and CYP epoxygenase pathway contribute to the neoplastic phenotype of MM cells.

Although a prominent feature of MM consists in the localization of MM cells in the bone marrow, a few MM cells can also be detected in the peripheral circulation. These observations suggest that MM cells have the capacity to circulate, invade and home to the bone marrow (Alsayed et al., 2007; Vande Broek et al., 2007). In the end stage of MM, the circulating plasma cells increase and grow at extramedullary sites (Pour et al., 2014). Clearly, the motility of MM cells is related to the disease progression. Numerous evidence has confirmed that EETs induce endothelial cell migration, even enhancing hematopoietic stem and progenitor cell homing and engraftment (Li et al., 2015). The present study showed that 17-ODYA can significantly inhibit, in a time-dependent manner, the motility of MM cells, including the migration and invasion of all tested MM cells. These results demonstrated that the epoxygenase pathway may be involved in the regulation of the motility of MM cells through the metabolites EETs. MM cells can localize in the bone marrow, which consists of extracellular matrix (ECM) and stromal cells, and MM cells have the capacity to invade and constitutively produce MMPs that are essential for matrix degradation (Barillé et al., 1997; Hecht et al., 2007; Vande Broek et al., 2004). In the present study, both U266 and RPMI 8226 cells expressed and secreted MMP-2 and MMP-9 proteins that can hydrolyze gelatin, but 17-ODYA reduced the activity of MMPs and suppressed the protein levels of both MMP-2 and MMP-9 in MM cells. These data revealed the role of the epoxygenase pathway in 17-ODYA-mediated inhibition of invasion and MMP protein levels.

Combined with our previous findings (Shao et al., 2011), the present study supports the involvement of the CYP epoxygenase pathway and the elevated levels of EETs in the proliferation, angiogenesis and motility of MM cells. We have identified the role of the third pathway of the arachidonic acid metabolism, the CYP epoxygenase pathway, in the pathogenesis of MM. It is not clear how active MM develops from the “dormancy period” of monoclonal gammopathy of undetermined significance (MGUS). Recently, EETs were found to stimulate tumor cells to escape from tumor dormancy in several tumor models (Panigrahy et al., 0000). Thus, we can reasonably hypothesize that EETs may promote MM to escape from this “dormancy period” and that the CYP epoxygenase pathway may be substantially involved in this progression. In conclusion, the CYP pathway could be an important therapeutic target, and the inhibitor 17-ODYA appears to be an attractive candidate for MM therapy.

Supplemental Information

Figure S1 The vehicle (DMSO) significantly inhibited the viability of MM cells

DMSO showed significant inhibitory effect on the viability of U266 (A) and RPMI 8226 (B) cells when used as a vehicle for EETs.

Click here for additional data file.

Data S1 Dataset

Click here for additional data file.

Additional Information and Declarations

Competing Interests

Author Contributions

Human Ethics

Data Availability

The authors declare there are no competing interests.

Jing Shao performed the experiments, analyzed the data, wrote the paper, prepared figures and/or tables, reviewed drafts of the paper.

Hongxiang Wang and Guolin Yuan performed the experiments, contributed reagents/materials/analysis tools.

Zhichao Chen conceived and designed the experiments.

Qiubai Li conceived and designed the experiments, analyzed the data, wrote the paper, reviewed drafts of the paper.

The following information was supplied relating to ethical approvals (i.e., approving body and any reference numbers):

Review Board of Tongji Medical College, Huazhong University of Science and Technology (No 2012S119)

The following information was supplied regarding data availability:

The raw data has been supplied as Data S1 .

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
