# Peer review of "Involvement of the arachidonic acid cytochrome P450 epoxygenase pathway in the proliferation and invasion of human multiple myeloma cells"

_PeerJ, doi:10.7717/peerj.1925_

## Round 0.1 · original submission · Major Revisions

See the comments from the reviewers. Some clarification is needed, but I feel you will be able to easily respond to the comments.

·

Basic reporting

No comments

Experimental design

Can the authors supply more information on the EET assays. How specific are the antibodies with regard cross reactivity to other EETs / DHETs? Are these EET/DHET assays? The authors need to be specific about what is being measured

The authors need clarification on the n-numbers used in each figure

in some experiments it would appear ANOVA may be more appropriate an analysis (e.g. figure 2).

Figure 2B – are EET levels normalised to account for any changes in cell viability

Figure 4 – the authors should perform densitometry and represent data additionally as a figure

Figure 5C and 5D – on the screen appear very dark and difficult to see.

It would be nice to see a break down of EET secretion from the different patient classes, compared to control.

Validity of the findings

no comments

Additional comments

This is a novel and interesting study that on the whole I believe just needs some clarification with regards methods and analysis used.

Reviewer 2 ·

Basic reporting

No comments

Experimental design

No comments

Validity of the findings

No comments

Additional comments

The paper described the role of the CYP epoxygenase pathway in cell proliferation, apoptosis, migration and invasion of MM. Generally, the method is well designed and the manuscript is clearly written.

I have only one recommendation regarding the correlation between EETs levels and prognostic factors that given as B2 microglobulin and LDH. Is it possible to give the cytogenetic and molecular features of the patients in table 1 whether the genetic risk factors have any effects on increased EETs levels? These simple findings may lead to results of the future studies.

Reviewer 3 ·

Basic reporting

The article may benefit from a review by an english professional

Experimental design

The experimental design is fine, although further experiments are needed, and well defined

Validity of the findings

Results are clear but must be described and discussed more cautiousely and considering what is enclosed in the General Comments

Additional comments

This is an interesting study on the effect of 17-ODYA and EETs in cell proliferation, apoptosis, migration and invasion of multiple myeloma (MM).
The Authors found that MM cell lines generated detectable levels of 11,12-EET/14,15-EET and that increased levels of EETs were found in the serum of MM patients compared to healthy donors. The addition of exogenous EETs induced significantly enhanced proliferation of MM cells, whereas 17-octadecynoic acid (17-ODYA), inhibited the viability and proliferation of MM cells. Moreover, this inhibitory effect could be successfully reversed by exogenous EETs. 17-ODYA also inhibited the motility of MM cells in a time-dependent manner, with a reduction of the gelatinolytic activity and protein expression of the matrix metalloproteinases (MMP)-2 and MMP-9.
The Authors conclude that the CYP epoxygenase pathway is involved in the proliferation and invasion of MM cells, for which 17-ODYA could be a promising therapeutic drug.

My major comment is that 17-ODYA is considered an inhibitor of cytochrome P-450 omega-hydroxylase (J Appl Physiol (1985). 2010 Aug;109(2):412-7; Circ Res. 2004 Oct 15;95(8):e65-71 etc) although it may be also be an epoxygenase inhibitor (J Cereb Blood Flow Metab. 2008 Aug;28(8):1431-9; Cancer Res. 2007 Jul 15;67(14):6665-74 etc): I think the Authors should discuss this point, even if they showed that 17-ODYA decreased EET (Fig. 2B), but did not consider 20-HETE. In a previous study (J Cereb Blood Flow Metab. 2008 Aug;28(8):1431-9): “17-ODYA did not inhibit the formation of EETs in tumor tissue, implying that 17-ODYA appears to exert its antitumorogenic function by a different mechanism that needs to be explored.”
I think it would be important to demonstrate that a different inhibitor of epoxygenase (miconazole, MS-PPOH) has the same effects as 17-ODYA, in order to conclude that EETs are important in the proliferation and invasion of MM cells. Maybe epoxygenase (which CYP450 in MM-cells?) could be measured.

Plasma levels of EETs are expressed in pg/ml: why not ng/ml?

Introduction: “Current evidence supports the notion of an association between deranged endogenous fatty acids and MM pathogenesis…”: can the Authors provide this “evidence”, which is not “evident” from ref. 3?

Methods: How did the Authors chose the amount of 17-ODYA used in the experiments, 100μmol/l? which is rather high.

Results: “We previously found secreted 11,12-EET and 14,15-EET in the supernatant of MM cell lines [15]. Therefore, we directly assessed the levels of 11,12-EET and 14,15-EET in MM cells.”: what does it mean?

The y axis of the second part of fig. 1B should read 14,15EET

Results: "17-ODYA suppressed EET levels and the proliferation of MM cells": I cannot see this “suppression”! In Fig. 2D only a very mild effect is evident.

Fig. 4: a semiquantitative evaluation of an adequate number of experiments is needed.

Discussion: “17-ODYA reduced the activity of MMPs and suppressed the protein levels of both MMP-2 and MMP-9 in MM cells..”: I cannot see this “suppression”.

---

## Round 0.2 · accepted · Accept

Thank you for your revision with appropriate responses.

·

Basic reporting

I'm happy with the authors changes to the manuscript and answers to the queries

Experimental design

I'm happy with the authors changes to the manuscript and answers to the queries

Validity of the findings

I'm happy with the authors changes to the manuscript and answers to the queries

Reviewer 3 ·

Basic reporting

Improved

Experimental design

The experimental design is fine.

Validity of the findings

Results are fine

Additional comments

I think the Authors have answered the questions and comments and the manuscript can be accepted for publication.